# Analysis of Mechanical Excavation Characteristics by Pre-Cutting Machine Based on Linear Cutting Tests

Han-eol Kim [1] , Sang-gui Ha [1], Hafeezur Rehman [2,3] and Han-kyu Yoo [1,*]

1 Department of Civil and Environmental System Engineering, Hanyang University, 55 Hanyangdaehak-ro, Sangnok-gu, Ansan 15588, Republic of Korea
2 Department of Mining Engineering, Balochistan University of Information Technology Engineering and Management Sciences (BUITEMS), Quetta 87300, Pakistan
3 School of Materials and Mineral Resources Engineering, Universiti Sains Malaysia, Nibong Tebal 14300, Penang, Malaysia
* Correspondence: hankyu@hanyang.ac.kr; Tel.: +82-31-400-5147

**Abstract:** Mechanical methods of tunnel excavation are widely used because of their high excavation output, and the selection of appropriate technology depends on ground composition and project-related features. Compared with tunnel boring machines (TBMs) and roadheaders, mechanical pre-cutting machines are used in tunnel widening and have proven to be reliable in tunnel capacity expansion. Compared to other machines, the excavation characteristics of pre-cutting machines are not systematically analyzed because of their rare use. In this study, the excavation characteristics of a pre-cutting machine are analyzed in a laboratory based on linear cutting tests performed on four rock specimens with different uniaxial compressive strengths. During testing, changes in tool forces, cutting volume, and specific energy are determined while maintaining different penetration depths, spacings, and rock strengths. The variations in these variables are selected accordingly. The results showed high similarity with the case of TBMs and roadheaders. However, in the excavation by the pre-cutting machine, the ratios of the peak-to-mean cutting forces and cutting-to-normal forces reached a maximum value at a specific $s/p$ (spacing and penetration ratio), which is related to the optimal cutting conditions. This study can provide useful information for the operation and design of pre-cutting machines.

**Keywords:** linear cutting test; pre-cutting machine; excavation machine; tool forces; cutting volume; specific energy

## 1. Introduction

Two common rock tunneling techniques are (1) mechanical method, and (2) drill and blast/conventional excavation methods. The primary differences between them are the excavation sequences, support installation, and construction progress. The conventional method has a wider range of applications in both new tunnels and enlarged existing tunnels [1]. The mechanical method includes a tunnel boring machine (TBM), roadheader, and pre-cutting machine. Selection of the excavation method depends on ground composition and project-related features, and the selected method directly affects the project schedule and cost [2]. TBMs and roadheaders are widely used in new tunnel construction; however, the pre-cutting machine has comparatively high applicability in the enlargement of existing tunnels. A pre-cutting machine is a tunnel excavation machine that excavates using a cutting tool, such as TBMs and roadheaders. In the operation of a tunnel excavation machine and design of the cutterhead, the physical and mechanical properties of the rock and excavation conditions, such as penetration depth and spacing, must be considered.

The pre-cutting method is a mechanical tunneling technique performed using a pre-cutting machine. In this method, a free surface is formed by pre-cutting along the edge of the planned tunnel cross section using an oversized chain-saw-shaped cutter head

(Figure 1). Because the effects of excavation and blasting cannot pass through a free surface [3,4], this technique does not disturb the surrounding ground, prevents ground settlement, and can accelerate tunneling [5]. This tunneling technique is economical, with a uniaxial compressive strength of approximately 70 MPa, and can be applied up to 100 MPa over short distances [6].

This tunneling technique was first introduced in the United States of America in 1950 and later rediscovered in France in 1970 and applied to the construction of transportation tunnels [6]. Incomplete statistics show that approximately 30 tunnels were built using this technique in France and Italy during the 1980s and 1990s [7]. In recent years, tunnel widening using pre-cutting machines has proven to be a good means of expanding the capacity of existing roads, highways, or railway tunnels while maintaining traffic flow [8,9] (Figure 1b). Therefore, the demand for mechanical precutting is expected to increase.

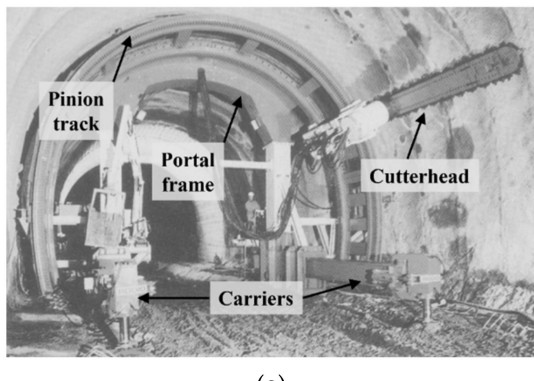
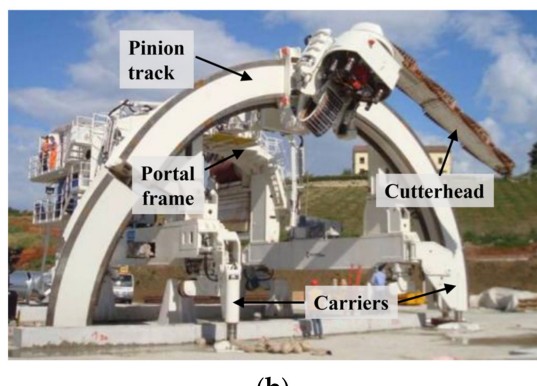

(**a**)                    (**b**)

**Figure 1.** Pre-cutting machine: (**a**) in a double-track railway tunnel (reproduced with permission from Ref. [6]. 1991. E. van Walsum); and (**b**) tunnel widening (reproduced with permission from Ref. [9]. 2014. Lunardi et al.).

Numerous studies have been conducted to provide useful information for the design and operation of TBMs and roadheaders. Snowdon et al. [10] analyzed the effects of excavation characteristics by penetration depth and spacing in cutting selected British rocks using TBM disc cutters. Sanio [11] investigated the excavation characteristics of a disk cutter for cutting anisotropic rock and provided useful performance predictions. Gertsch et al. [12] analyzed the excavation characteristics to predict the performance of a TBM from the cutting test of Colorado red granite. Balci et al. [13] investigated the relationship between optimal specific energy and rock properties in a roadheader using linear cutting tests. Bilgin et al. [14] investigated the dominant rock properties that affect the performance of conical picks using rock-cutting tests. Tiryaki and Dikmen [15] analyzed the effect of rock properties on the specific energy during cutting using pick cutters. Wang et al. [16] analyzed the effect of penetration depth and spacing on the cutting performance when cutting using conical picks. Yasar and Yilmaz [17] provided useful information after analyzing the mechanical excavation characteristics based on cutting conditions during cutting using a chisel pick. Özşen et al. [18] investigated the relationship between the specific energy and physical and mechanical properties of rocks excavated by roadheaders. Huang et al. [19] analyzed the excavation characteristics of granite under conical picks using indentation tests. However, the excavation characteristics of the pre-cutting machines were not analyzed compared to TBMs and roadheaders. As the demand for pre-cutting machines is expected to increase, it is essential to analyze the mechanical excavation characteristics of these machines.

In this study, linear cutting tests were performed in the laboratory using a cutting tool of a precutting machine. Tests were performed on four rock specimens with different uniaxial compressive strengths. The purpose of this study is to analyze the mechanical

excavation characteristics of a pre-cutting machine and provide useful information for the operation and design of this machine.

## 2. Experimental Setup and Procedures

### 2.1. Cutting Tool and Rock Specimens

Cutting tools are classified as drag-type or roller-type tools. The conical pick of roadheaders is an example of a drag type, and this type of cutting tool is usually used in a partial-face excavation machine. Moreover, a TBM disc cutter is an example of a roller type tool, and it is usually used in a full-face excavation machine [20]. A pre-cutting machine can be classified as a partial-face machine; therefore, a drag-type tool is used.

The shape of the cutting tool used in this study was obtained from a previous work [21]. The specifications of the cutting tool are shown in Figure 2a. The length (h), thickness (t), and width (d) were 80, 30, and 60 mm, respectively. The edge angle ($\theta$), clearance angle ($\alpha$) and rake angle ($\beta$) were 120°, 10° and 5°, respectively. This tool was manufactured using SKD 11 alloy steel with a hardness greater than 60 HRC to reduce tool wear. Moreover, the edge in contact with the rock was filled to a radius of 2 mm. Its actual appearance is shown in Figure 2b.

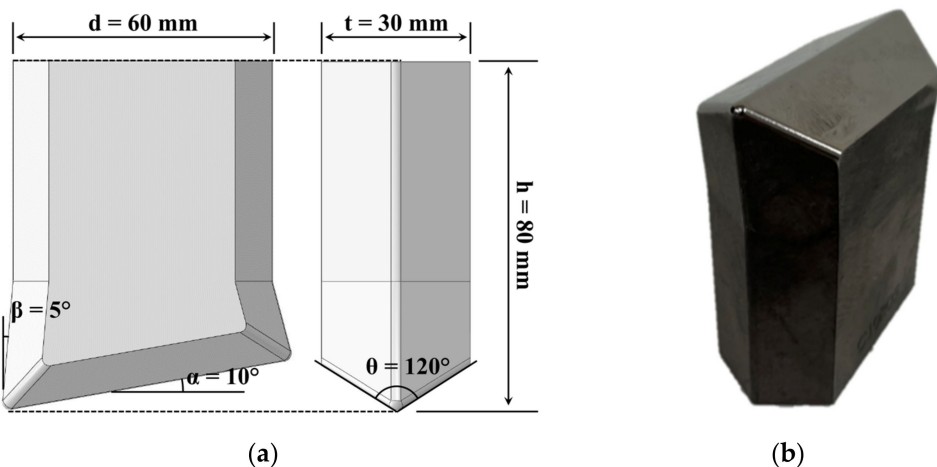

**Figure 2.** Cutting tool used in the experiment: (**a**) specification; (**b**) appearance.

It is difficult to obtain rocks of the same composition and strength. To overcome this limitation and for simplicity, rock-like materials such as ceramics and concrete have been used as substitutes in rock experiments [22–24]. In this study, rock specimens were produced using mortar consisting of sand and cement.

The rock specimens were manufactured with dimensions 400 mm × 400 mm × 300 mm. Moreover, because mechanical precutting is effective when the compressive strength of the rock is less than 70 MPa [6], the strength of the rock specimens was designed to be less than 50 MPa. After a curing period of 28 days, uniaxial compression and Brazilian tests were performed to verify that the strengths of the rock specimens reached the target value (Figure 3). The target rock strengths and the physical and mechanical properties obtained from the tests are listed in Table 1.

### 2.2. Linear Cutting Machine

The linear cutting test is known as the most reliable method for the design and performance prediction of mechanical excavation machines [25]. Linear cutting machines can be classified into full-scale and small-scale linear cutting machines. Since the full-scale cutting machine uses a large sample, the test result can be directly used in an excavation machine [26]. Small-scale linear cutting machines were developed before full-scale machines [27]. Rock samples from small-scale linear cutting machines have the advantage of

being easily and inexpensively obtained from core or rectangular block samples, obtained during a geological investigation [28].

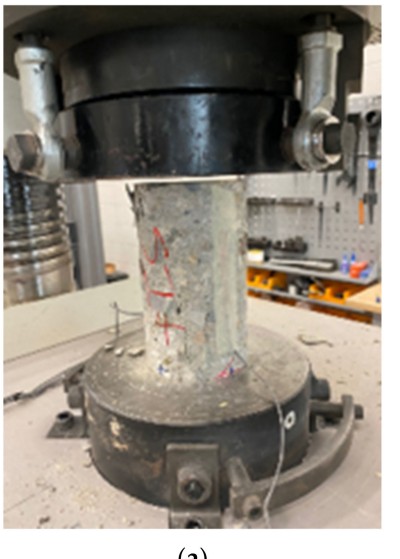 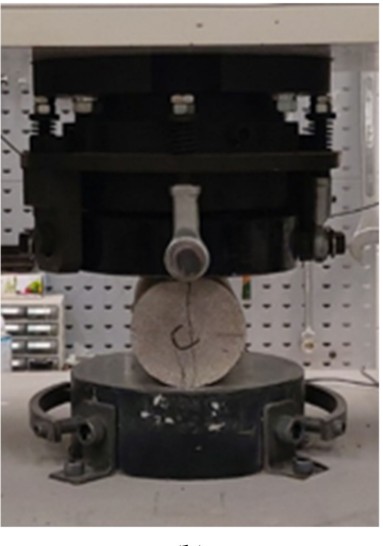

(**a**)　　　　　　　　　　　　　　　　(**b**)

**Figure 3.** Tests to obtain the mechanical properties of rock specimens: (**a**) uniaxial compression and (**b**) Brazilian tests.

**Table 1.** Physical and mechanical properties of rock specimens.

| Target Strength (MPa) | Elastic Modulus (Gpa) | Density (kg/m³) | Poisson's Ratio | Uniaxial Compressive Strength, UCS (Mpa) | Brazilian Tensile Strength, BTS (Mpa) |
|---|---|---|---|---|---|
| 20 | 16.92 | 2214 | 0.3 | 18.0 | 2.06 |
| 30 | 33.35 | 2363 | 0.3 | 29.3 | 2.18 |
| 40 | 38.92 | 2382 | 0.3 | 42.0 | 2.51 |
| 50 | 44.47 | 2235 | 0.3 | 51.8 | 2.99 |

In this study, rock-cutting tests were performed using a small-scale linear cutting machine, as shown in Figure 4a. A linear cutting machine is divided into cutting and control systems. The cutting system includes a frame of the main body, bucket for holding rocks, driving device for a machine, and load cells. The control system consists of a computer and monitor.

The linear cutting machine used in this study is driven by a servomotor rather than a commonly used hydraulic cylinder. The servomotor has the advantage of precision control in terms of the moving distance. The servomotor in the x-direction has a stroke of 500 mm in the left and right directions and is used to set the cutting spacing. The y-direction servomotor has a stroke of 700 mm in the front-rear direction and is used for cutting the rocks (Figure 4b). A z-direction servomotor is used to set the penetration depth. The servomotors in the x- and y-directions used a variable type, with a maximum speed of 100 mm/s.

In general, the tool forces generated in three directions are measured using a triaxial load cell. However, in this study, three uniaxial load cells were installed in each direction to accurately measure tool forces. Because the side force ($F_s$) generated in the x-direction is relatively small, it was measured using a 15-ton load cell. The cutting force ($F_c$) and normal force ($F_n$) generated in the y- and z-directions, respectively, were measured using 25-ton load cells (Figure 4c). Because one cutting action is completed in a short time during a linear cutting test, a load cell capable of accumulating data at 50 Hz is used to increase the amount of available data and the reliability of the test results.

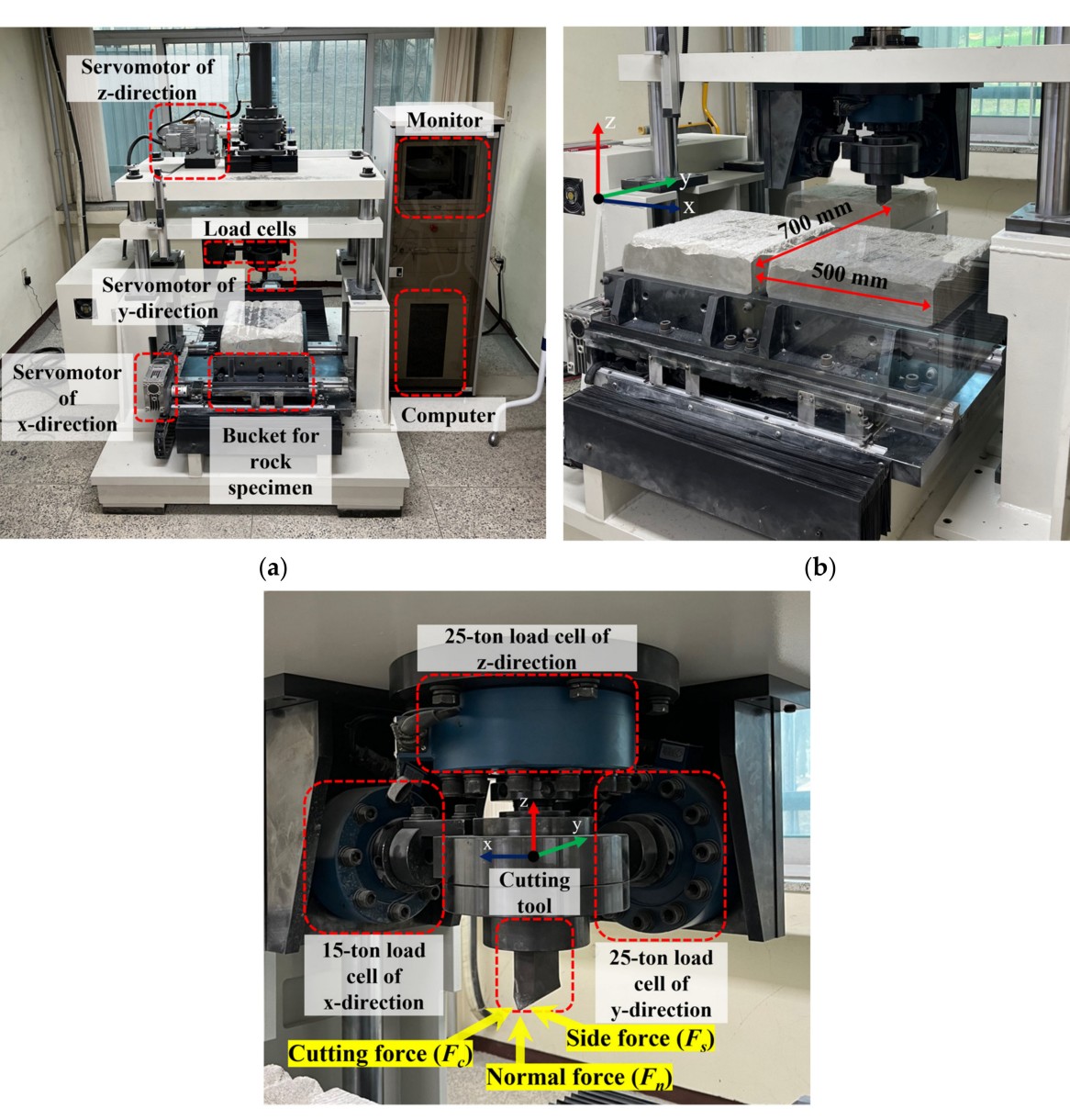

**Figure 4.** Linear cutting machine (LCM): (**a**) overview of LCM, (**b**) stroke of servomotor in x- and y-direction, and (**c**) load cells and cutting tool.

The control system may set cutting conditions such as the penetration depth and spacing. As the cutting length, penetration depth, spacing, and number of cuttings were input, a series of cuttings were automatically performed. The tool forces generated during cutting were displayed in real time via the monitor. In addition, the speed of the servomotors and data measurement period of the load cell can be set.

### 2.3. Cutting Scheme

Before the main cutting, a series of cuttings is performed twice or thrice. These series of cuts are defined as pre-conditioning. Because the tunnel excavation machine excavates the damaged face in the field, pre-conditioning is performed to simulate conditions similar to the actual excavation site in the linear cutting test (Figure 5).

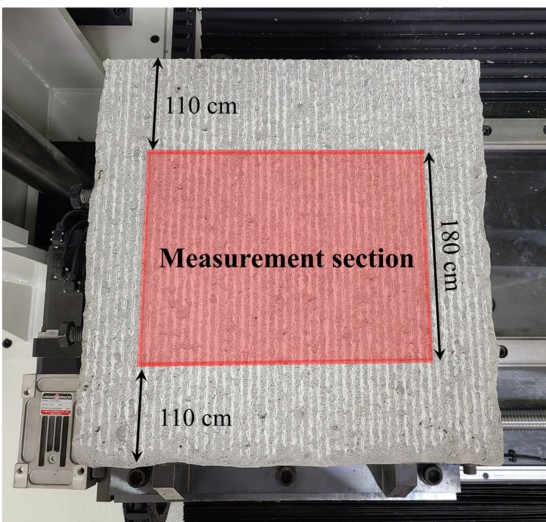

**Figure 5.** Rock surface after pre-conditioning and measurement section.

The main cutting tests were performed using penetration depths of 3, 6, and 9 mm; the spacings used for each penetration depth are listed in Table 2. Because the cutting speed does not affect the cutting performance [29], it was arbitrarily set to 12.5 mm/s. The cutting test consisted of five cuts. Rock fragments and tool forces were used only from the measurement area in Figure 5, and the results from the first cut were excluded from the analysis because they did not represent an interaction between the cuts.

**Table 2.** Penetration depth and spacing used in cutting tests.

| Penetration Depth, $p$ (mm) | Spacing, $s$ (mm) | | | | | | | | | |
|---|---|---|---|---|---|---|---|---|---|---|
| 3 | 3 | 6 | 9 | 12 | 15 | 18 | 21 | 24 | 27 | 30 |
| 6 | 4 | 8 | 12 | 16 | 20 | 24 | 28 | 32 | | |
| 9 | 8 | 16 | 24 | 32 | 40 | 48 | | | | |

The specific energy (*SE*) represents the excavation efficiency of the tunnel excavation machine, which indicates the energy consumed to excavate the unit volume. The excavation efficiency is maximized when the specific energy is minimized, and this energy is obtained using Equation (1).

$$SE = \frac{F_c \times l}{V_c} \tag{1}$$

where $F_c$ is the mean cutting force, $l$ is the length of the cut, and $V_c$ is the cutting volume. The cutting force ($F_c$) is the average of the second to fifth cuts measured in the measurement area, and the cutting volume ($V_c$) was obtained through 3D scanning.

## 3. Results

### 3.1. Analysis of Characteristics of Tool Forces by Cutting Conditions

3.1.1. Effect of Penetration Depth and Spacing on Normal Force

A normal force is generated in the direction perpendicular to the cutting direction and cutting plane. It is used to estimate the effective mass and thrust of the excavation machine and maintain the desired depth of penetration [29]. The normal force increases with the penetration depth and spacing in cutting discs [11,12,30–33], and the same is true for cutting with a pick cutter [14,16,34].

The variations In the mean normal force with penetration depth and spacing are shown in Figure 6 for all rock specimens, and the details are listed in Tables 3–6. The normal force in cutting with the tool of the pre-cutting machine increased linearly as the penetration depth increased. The increase in the spacing also increased the normal force

and formed a linear relationship. However, the correlation with the spacing ($R^2 > 0.602$) was stronger than that with the penetration depth ($R^2 < 0.587$). This indicates that the spacing affects the normal force more than the penetration depth.

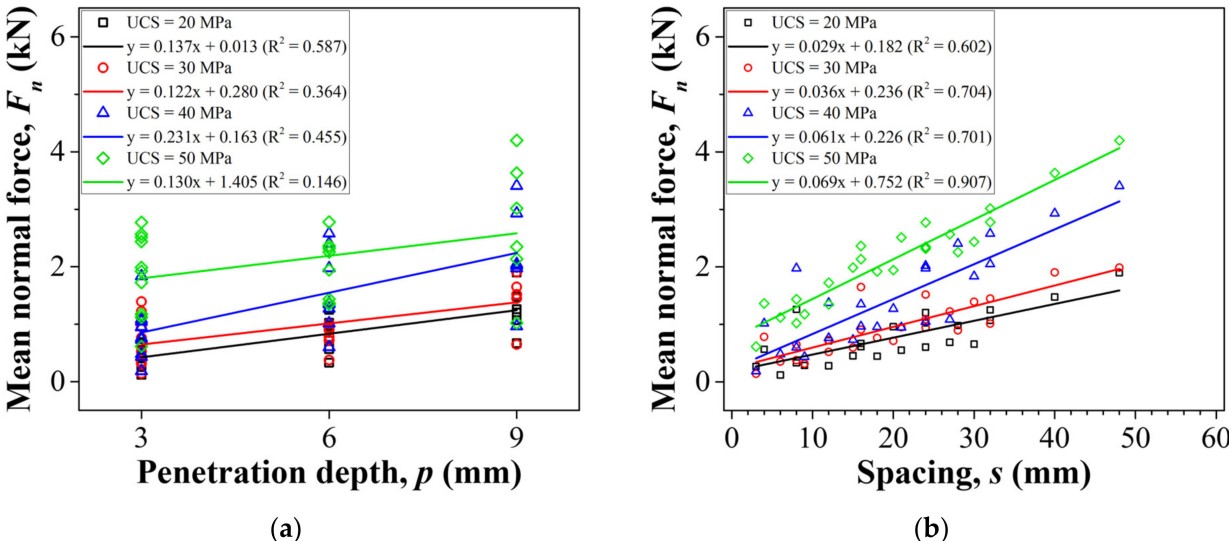

**Figure 6.** Relationship between the mean normal force and (**a**) penetration depth, (**b**) spacing.

**Table 3.** Rock cutting results of UCS 20 MPa.

| $p$ (mm) | $s$ (mm) | $s/p$ | $F_n$ (kN) | $F'_n$ (kN) | $F'_n/F_n$ | $F_c$ (kN) | $F'_c$ (kN) | $F'_c/F_c$ | $F_c/F_n$ |
|---|---|---|---|---|---|---|---|---|---|
| 3 | 3 | 1 | 0.27 | 0.61 | 2.28 | 0.36 | 0.88 | 2.41 | 1.44 |
| | 6 | 2 | 0.12 | 0.34 | 2.86 | 0.38 | 1.08 | 2.86 | 3.21 |
| | 9 | 3 | 0.28 | 0.78 | 2.73 | 0.40 | 1.14 | 2.83 | 1.47 |
| | 12 | 4 | 0.28 | 0.67 | 2.41 | 0.55 | 1.46 | 2.65 | 2.16 |
| | 15 | 5 | 0.45 | 1.10 | 2.42 | 0.85 | 2.02 | 2.38 | 1.84 |
| | 18 | 6 | 0.45 | 1.18 | 2.65 | 0.82 | 2.22 | 2.70 | 1.88 |
| | 21 | 7 | 0.55 | 1.10 | 1.99 | 0.98 | 2.05 | 2.09 | 1.87 |
| | 24 | 8 | 0.60 | 1.29 | 2.13 | 1.02 | 2.28 | 2.25 | 1.77 |
| | 27 | 9 | 0.69 | 1.46 | 2.12 | 1.15 | 2.60 | 2.26 | 1.78 |
| | 30 | 10 | 0.65 | 1.43 | 2.18 | 1.09 | 2.41 | 2.20 | 1.69 |
| 6 | 4 | 0.67 | 0.57 | 1.26 | 2.22 | 0.83 | 1.93 | 2.32 | 1.53 |
| | 8 | 1.33 | 0.33 | 0.76 | 2.31 | 0.99 | 2.57 | 2.60 | 3.37 |
| | 12 | 2 | 0.71 | 1.62 | 2.27 | 1.22 | 3.05 | 2.50 | 1.88 |
| | 16 | 2.67 | 0.61 | 1.57 | 2.57 | 1.37 | 3.83 | 2.80 | 2.43 |
| | 20 | 3.33 | 0.96 | 2.24 | 2.34 | 1.66 | 4.25 | 2.57 | 1.89 |
| | 24 | 4 | 1.02 | 2.32 | 2.27 | 1.82 | 5.22 | 2.88 | 2.25 |
| | 28 | 4.67 | 0.98 | 2.33 | 2.38 | 1.83 | 4.97 | 2.71 | 2.14 |
| | 32 | 5.33 | 1.25 | 2.57 | 2.06 | 2.27 | 5.31 | 2.34 | 2.07 |
| 9 | 8 | 0.89 | 1.26 | 2.46 | 1.95 | 1.99 | 4.52 | 2.27 | 1.84 |
| | 16 | 1.78 | 0.67 | 1.46 | 2.19 | 2.07 | 5.44 | 2.63 | 3.72 |
| | 24 | 2.67 | 1.20 | 3.04 | 2.53 | 2.53 | 7.74 | 3.06 | 2.55 |
| | 32 | 3.56 | 1.07 | 2.48 | 2.33 | 2.73 | 7.50 | 2.75 | 3.02 |
| | 40 | 4.44 | 1.48 | 3.03 | 2.05 | 3.24 | 8.30 | 2.56 | 2.74 |
| | 48 | 5.33 | 1.90 | 3.80 | 2.00 | 3.29 | 8.92 | 2.71 | 2.35 |

$p$: penetration depth, $s$: spacing, $F_n$: mean normal force, $F'_n$: peak normal force, $F_c$: mean cutting force, $F'_c$: peak cutting force.

**Table 4.** Rock cutting results of UCS 30 MPa.

| $p$ (mm) | $s$ (mm) | $s/p$ | $F_n$ (kN) | $F'_n$ (kN) | $F'_n/F_n$ | $F_c$ (kN) | $F'_c$ (kN) | $F'_c/F_c$ | $F_c/F_n$ |
|---|---|---|---|---|---|---|---|---|---|
| 3 | 3 | 1 | 0.14 | 0.28 | 2.00 | 0.39 | 1.13 | 2.87 | 4.04 |
| | 6 | 2 | 0.35 | 1.30 | 3.71 | 0.58 | 1.75 | 3.02 | 1.35 |
| | 9 | 3 | 0.31 | 1.06 | 3.38 | 0.80 | 2.26 | 2.82 | 2.13 |
| | 12 | 4 | 0.52 | 1.98 | 3.81 | 0.87 | 2.41 | 2.77 | 1.22 |
| | 15 | 5 | 0.60 | 1.77 | 2.98 | 1.02 | 2.75 | 2.70 | 1.55 |
| | 18 | 6 | 0.76 | 2.19 | 2.87 | 1.31 | 3.37 | 2.57 | 1.54 |
| | 21 | 7 | 0.95 | 2.67 | 2.82 | 1.38 | 3.41 | 2.48 | 1.28 |
| | 24 | 8 | 1.09 | 2.78 | 2.55 | 1.53 | 3.74 | 2.44 | 1.35 |
| | 27 | 9 | 1.22 | 2.62 | 2.14 | 1.70 | 3.87 | 2.27 | 1.47 |
| | 30 | 10 | 1.39 | 2.88 | 2.07 | 1.91 | 4.21 | 2.20 | 1.46 |
| 6 | 4 | 0.67 | 0.78 | 1.47 | 1.87 | 0.74 | 1.87 | 2.53 | 1.27 |
| | 8 | 1.33 | 0.38 | 0.78 | 2.08 | 0.79 | 2.52 | 3.20 | 3.22 |
| | 12 | 2 | 0.72 | 1.69 | 2.33 | 0.87 | 2.92 | 3.34 | 1.73 |
| | 16 | 2.67 | 0.91 | 1.90 | 2.09 | 1.58 | 4.05 | 2.56 | 2.13 |
| | 20 | 3.33 | 0.71 | 2.40 | 3.36 | 1.62 | 4.20 | 2.59 | 1.75 |
| | 24 | 4 | 0.94 | 2.97 | 3.15 | 1.84 | 4.76 | 2.59 | 1.61 |
| | 28 | 4.67 | 0.89 | 2.17 | 2.42 | 1.90 | 4.87 | 2.56 | 2.25 |
| | 32 | 5.33 | 1.01 | 2.59 | 2.56 | 2.08 | 5.47 | 2.64 | 2.11 |
| 9 | 8 | 0.89 | 0.65 | 1.43 | 2.21 | 1.76 | 4.93 | 2.80 | 3.45 |
| | 16 | 1.78 | 1.65 | 3.79 | 2.30 | 2.37 | 6.71 | 2.83 | 1.77 |
| | 24 | 2.67 | 1.52 | 3.49 | 2.30 | 2.31 | 6.89 | 2.98 | 1.98 |
| | 32 | 3.56 | 1.45 | 3.42 | 2.36 | 3.07 | 8.64 | 2.82 | 2.53 |
| | 40 | 4.44 | 1.90 | 4.20 | 2.20 | 3.44 | 9.44 | 2.74 | 2.25 |
| | 48 | 5.33 | 1.99 | 4.41 | 2.22 | 3.34 | 8.40 | 2.52 | 1.91 |

$p$: penetration depth, $s$: spacing, $F_n$: mean normal force, $F'_n$: peak normal force, $F_c$: mean cutting force, $F'_c$: peak cutting force.

**Table 5.** Rock cutting results of UCS 40 MPa.

| $p$ (mm) | $s$ (mm) | $s/p$ | $F_n$ (kN) | $F'_n$ (kN) | $F'_n/F_n$ | $F_c$ (kN) | $F'_c$ (kN) | $F'_c/F_c$ | $F_c/F_n$ |
|---|---|---|---|---|---|---|---|---|---|
| 3 | 3 | 1 | 0.19 | 0.43 | 2.33 | 0.63 | 1.29 | 2.06 | 2.97 |
| | 6 | 2 | 0.48 | 1.11 | 2.30 | 0.59 | 1.52 | 2.60 | 1.37 |
| | 9 | 3 | 0.43 | 1.06 | 2.45 | 0.69 | 1.86 | 2.68 | 1.75 |
| | 12 | 4 | 0.76 | 1.63 | 2.14 | 1.04 | 2.47 | 2.38 | 1.51 |
| | 15 | 5 | 0.73 | 1.65 | 2.27 | 1.13 | 2.59 | 2.30 | 1.57 |
| | 18 | 6 | 0.95 | 2.02 | 2.12 | 1.25 | 2.84 | 2.27 | 1.40 |
| | 21 | 7 | 0.95 | 1.91 | 2.02 | 1.08 | 2.48 | 2.29 | 1.30 |
| | 24 | 8 | 1.05 | 2.15 | 2.06 | 1.12 | 2.59 | 2.31 | 1.21 |
| | 27 | 9 | 1.08 | 2.74 | 2.53 | 1.44 | 3.40 | 2.36 | 1.24 |
| | 30 | 10 | 1.84 | 3.12 | 1.70 | 1.95 | 3.54 | 1.82 | 1.14 |
| 6 | 4 | 0.67 | 1.02 | 2.10 | 2.07 | 1.23 | 2.73 | 2.22 | 1.30 |
| | 8 | 1.33 | 0.60 | 1.30 | 2.16 | 1.37 | 3.49 | 2.55 | 2.70 |
| | 12 | 2 | 1.37 | 2.91 | 2.12 | 1.81 | 4.68 | 2.59 | 1.61 |
| | 16 | 2.67 | 1.35 | 2.82 | 2.09 | 1.89 | 4.66 | 2.47 | 1.65 |
| | 20 | 3.33 | 1.27 | 3.06 | 2.41 | 2.07 | 5.54 | 2.68 | 1.81 |
| | 24 | 4 | 1.98 | 4.38 | 2.22 | 2.58 | 6.96 | 2.70 | 1.59 |
| | 28 | 4.67 | 2.40 | 4.39 | 1.83 | 3.05 | 6.80 | 2.23 | 1.55 |
| | 32 | 5.33 | 2.58 | 4.79 | 1.86 | 3.16 | 6.98 | 2.21 | 1.46 |
| 9 | 8 | 0.89 | 1.98 | 3.86 | 1.95 | 2.63 | 6.07 | 2.31 | 1.57 |
| | 16 | 1.78 | 0.96 | 2.52 | 2.61 | 2.23 | 6.62 | 2.97 | 2.63 |
| | 24 | 2.67 | 2.02 | 4.12 | 2.04 | 2.96 | 8.47 | 2.86 | 2.06 |
| | 32 | 3.56 | 2.05 | 4.39 | 2.15 | 3.74 | 8.95 | 2.40 | 2.04 |
| | 40 | 4.44 | 2.93 | 5.31 | 1.81 | 4.30 | 10.19 | 2.37 | 1.92 |
| | 48 | 5.33 | 3.41 | 5.85 | 1.72 | 5.14 | 11.39 | 2.22 | 1.95 |

$p$: penetration depth, $s$: spacing, $F_n$: mean normal force, $F'_n$: peak normal force, $F_c$: mean cutting force, $F'_c$: peak cutting force.

**Table 6.** Rock cutting results of UCS 50 MPa.

| $p$ (mm) | $s$ (mm) | $s/p$ | $F_n$ (kN) | $F'_n$ (kN) | $F'_n/F_n$ | $F_c$ (kN) | $F'_c$ (kN) | $F'_c/F_c$ | $F_c/F_n$ |
|---|---|---|---|---|---|---|---|---|---|
| 3 | 3 | 1 | 0.61 | 1.53 | 2.49 | 0.58 | 1.89 | 3.28 | 0.58 |
| | 6 | 2 | 1.12 | 2.35 | 2.10 | 1.26 | 3.30 | 2.62 | 1.26 |
| | 9 | 3 | 1.18 | 2.93 | 2.49 | 1.67 | 4.27 | 2.56 | 1.67 |
| | 12 | 4 | 1.72 | 3.59 | 2.08 | 2.11 | 4.95 | 2.34 | 2.11 |
| | 15 | 5 | 1.99 | 4.23 | 2.13 | 2.61 | 6.43 | 2.46 | 2.61 |
| | 18 | 6 | 1.92 | 4.62 | 2.40 | 2.86 | 7.12 | 2.49 | 2.86 |
| | 21 | 7 | 2.51 | 4.69 | 1.87 | 3.00 | 5.99 | 2.00 | 3.00 |
| | 24 | 8 | 2.77 | 5.54 | 2.00 | 3.28 | 6.81 | 2.07 | 3.28 |
| | 27 | 9 | 2.57 | 4.65 | 1.81 | 3.12 | 5.89 | 1.89 | 3.12 |
| | 30 | 10 | 2.44 | 5.23 | 2.15 | 2.94 | 6.22 | 2.12 | 2.94 |
| 6 | 4 | 0.67 | 1.36 | 2.80 | 2.05 | 1.48 | 3.60 | 2.43 | 1.48 |
| | 8 | 1.33 | 1.44 | 2.78 | 1.93 | 2.03 | 5.33 | 2.63 | 2.03 |
| | 12 | 2 | 1.35 | 4.27 | 3.17 | 2.43 | 6.90 | 2.84 | 2.43 |
| | 16 | 2.67 | 2.36 | 5.49 | 2.32 | 2.61 | 7.82 | 3.00 | 2.61 |
| | 20 | 3.33 | 1.94 | 4.75 | 2.44 | 2.57 | 7.56 | 2.94 | 2.57 |
| | 24 | 4 | 2.32 | 5.37 | 2.32 | 3.18 | 8.96 | 2.81 | 3.18 |
| | 28 | 4.67 | 2.26 | 5.06 | 2.25 | 3.52 | 8.99 | 2.56 | 3.52 |
| | 32 | 5.33 | 2.78 | 5.78 | 2.08 | 3.90 | 10.15 | 2.60 | 3.90 |
| 9 | 8 | 0.89 | 1.02 | 2.40 | 2.36 | 3.34 | 7.76 | 2.32 | 3.34 |
| | 16 | 1.78 | 2.13 | 4.84 | 2.27 | 3.43 | 10.56 | 3.08 | 3.43 |
| | 24 | 2.67 | 2.35 | 5.56 | 2.37 | 4.62 | 12.80 | 2.77 | 4.62 |
| | 32 | 3.56 | 3.02 | 6.86 | 2.28 | 4.84 | 14.16 | 2.93 | 4.84 |
| | 40 | 4.44 | 3.63 | 8.02 | 2.21 | 7.08 | 17.65 | 2.50 | 7.08 |
| | 48 | 5.33 | 4.20 | 7.95 | 1.89 | 7.28 | 14.32 | 1.97 | 7.28 |

$p$: penetration depth, $s$: spacing, $F_n$: mean normal force, $F'_n$: peak normal force, $F_c$: mean cutting force, $F'_c$: peak cutting force.

### 3.1.2. Effect of Penetration Depth and Spacing on Cutting Force

The cutting force was generated parallel to the cutting plane and the cutting direction. It is directly related to the torque requirement of the excavation machine and is used to calculate the specific energy [29]. Several studies have reported that the cutting force increases as the penetration depth and spacing increase, regardless of the type of cutting tool used [12,14,16,17,30–32,34].

The variations in the mean cutting force with penetration depth and spacing are shown in Figure 7 for all rock specimens, and the details are listed in Tables 3–6. When the pre-cutting machine tool is used, the cutting force increased linearly with penetration depth and spacing. In addition, the cutting force, similar to the normal force, was influenced more by the spacing than the penetration depth.

### 3.1.3. Analysis of Peak to Mean Tool Forces Ratio

The ratio of peak-to-mean tool forces is an important factor in the design of machines and cutting tools. This ratio affects the vibration of the cutting head and breakdown of the mechanical parts. Machine vibrations generally increase as the ratio increases [29]. Considering that the ratio of the peak to mean tool forces does not correlate with the penetration depth [35], the relationship with the $s/p$ ratio was analyzed in this study.

Figure 8a shows the relationship between the ratio of the peak normal force to the mean normal force and $s/p$ ratio. This ratio was between approximately 1.6 and 3.8. However, it did not form a specific correlation with the $s/p$ ratio regardless of the rock specimen strength.

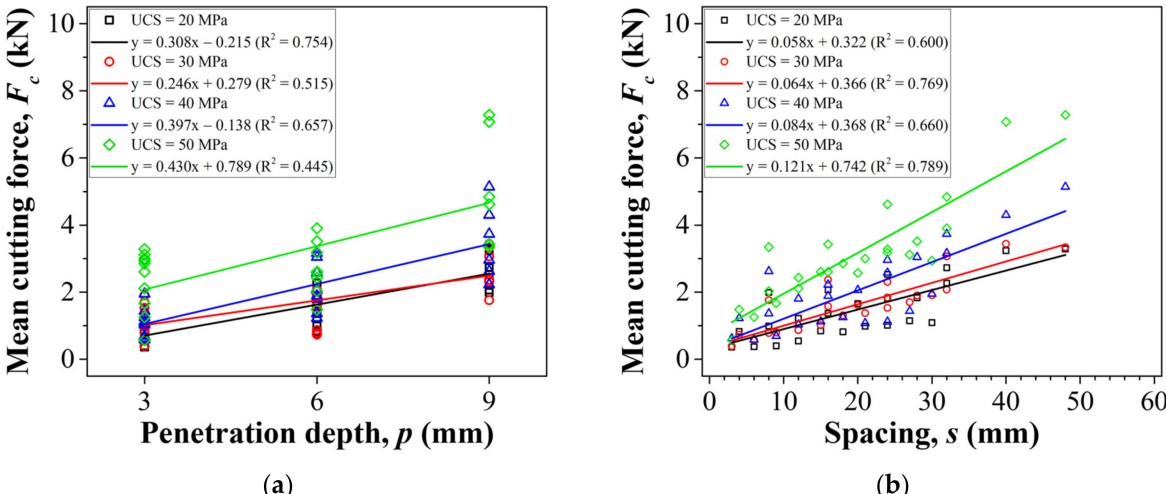

**Figure 7.** Relationship between mean cutting force and (**a**) penetration depth, (**b**) spacing.

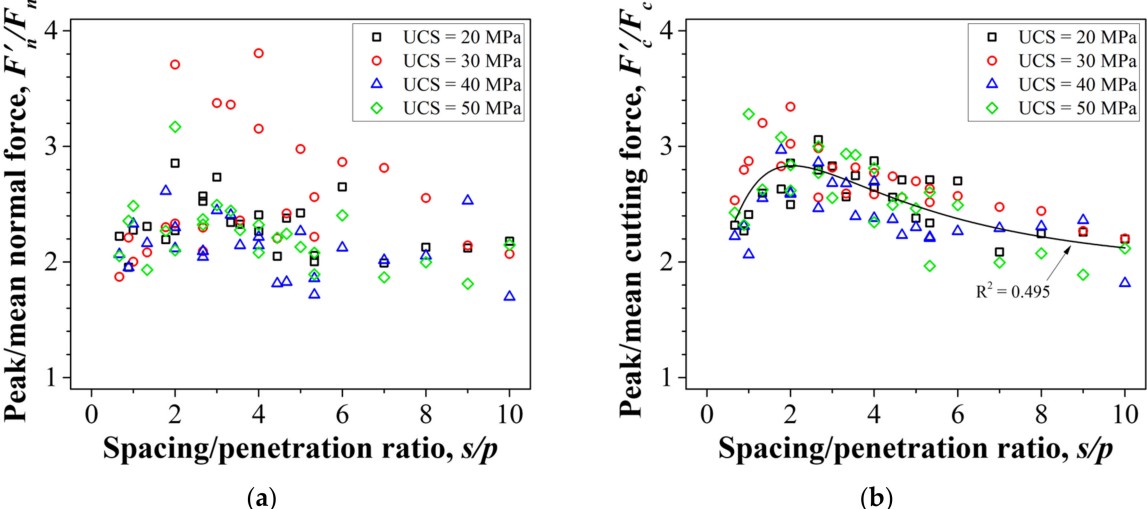

**Figure 8.** Relationships between $s/p$ ratio and ratio of peak to mean tool forces: (**a**) normal force and (**b**) cutting force.

The relationship between the ratio of the peak cutting force to the mean cutting force and $s/p$ ratio is shown in Figure 8b. This ratio was found to be between approximately 1.8 and 3.4, regardless of the rock strength. This result was consistent with that of Bilgin et al. [14] that the ratio of peak to mean tool forces was not affected by rock properties. The ratio of the peak to mean cutting force had a relatively significant correlation with the $s/p$ ratio ($R^2 = 0.495$). This ratio is maximized when $s/p$ is close to 2, regardless of the strength. Bilgin et al. [14] reported that the larger the ratio of the peak to mean cutting force, the larger the rock chips obtained. Therefore, in excavation using a pre-cutting machine, larger rock chips can be expected as the ratio of the spacing to the penetration depth approaches approximately 2. The details of the peak-to-mean tool force ratios are listed in Tables 3–6.

### 3.1.4. Analysis of Peak to Mean Tool Forces Ratio

The ratio of the cutting force to the normal force is an essential factor in the efficiency of machine excavation. In the TBM, this ratio is expressed as a cutting coefficient, which is considered an indicator of the amount of torque required for a given amount of thrust [36]. This ratio increases with increasing penetration depth and spacing during disc

cutters [12,30,32]. In cutting with drag-type tools, the ratio of the cutting to the normal force is affected by the wear of the tool. The normal force increases rapidly compared to the cutting force owing to tool wear, and this ratio decreases [27]. Even in a pick cutter, this ratio increases as the penetration depth and spacing increase [16].

The relationship between the ratio of the cutting and normal forces and the penetration depth is shown in Figure 9a. Regardless of the rock strength, the ratio of the cutting to normal force and penetration depth formed a positive linear relationship. However, except for USC 50 MPa ($R^2$ = 0.425), the correlation was weaker ($R^2$ < 0.08). It can be predicted that this ratio is not significantly affected by the penetration depth during the cutting action by the precutting machine.

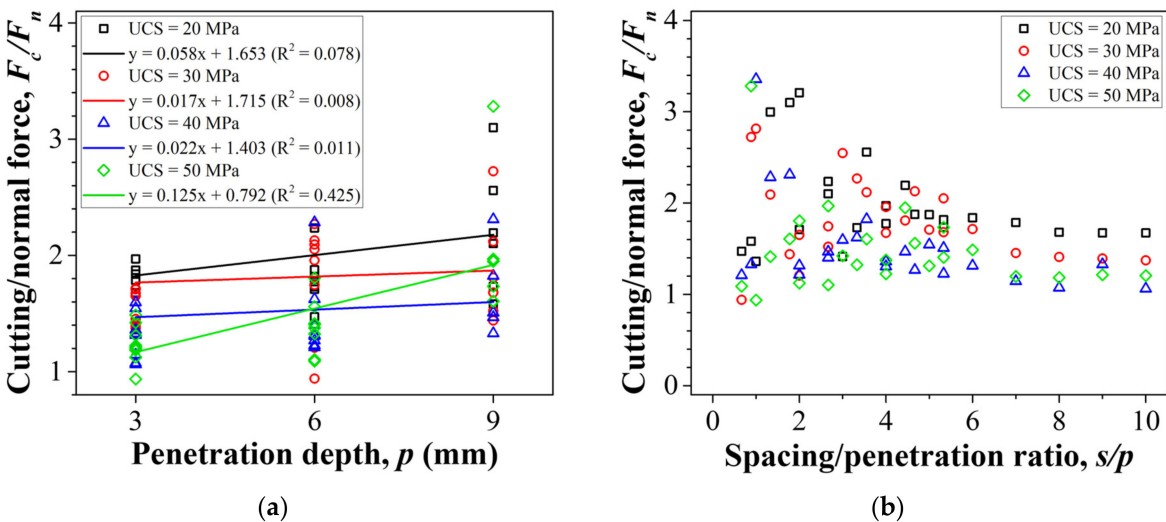

**Figure 9.** Relationships between ratio of cutting to normal force and (**a**) penetration depth, (**b**) $s/p$ ratio.

Figure 9b shows the relationship between the ratio of the cutting force to the normal force and $s/p$ ratio. The ratio of the cutting to normal force had no correlation with the spacing. However, it is noteworthy that this ratio reaches its maximum value when $s/p$ is close to 2 in relation to the $s/p$ ratio. The maximum value of this ratio at a UCS of 20 MPa was approximately 3.3; however, it was approximately 2.0 at a UCS of 50 MPa. This was probable because the wear of the cutting tool was lowest when $s/p$ was approximately 2, and the tool wear increased as the rock strength increased. The details of the ratios of the cutting and normal forces are listed in Tables 3–6.

### 3.2. Analysis of Characteristics of Cutting Volume by Cutting Conditions

3.2.1. Effect of Penetration Depth on Cutting Volume

The cutting volume represents the total volume of the rock fragments generated by cutting. Because the cutting volume is directly related to the calculation of the specific energy, the analysis of the effect of the cutting volume on the cutting condition is an important factor. In this study, the cutting volume was obtained by 3D scanning.

Figure 10 shows the relationship between the penetration depth and cutting volume for all rock specimens. Bilgin [37] reported that the cutting volume of a disc cutter increases exponentially as the penetration depth increases. Yasar and Yilmaz [20] reported that the cutting volume forms a power function relationship with the penetration depth during cutting by a pick cutter. In this study, the cutting volume increased within the power function relationship as the penetration depth increased, regardless of rock strength. In addition, the cutting volume was not affected by the rock strength.

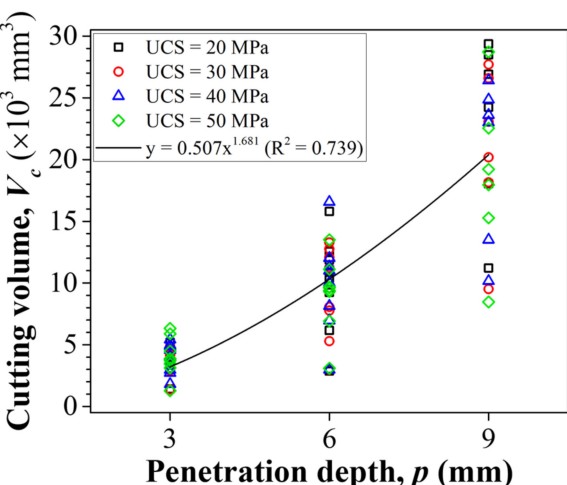

**Figure 10.** Relationship between cutting volume and penetration depth.

### 3.2.2. Effect of Penetration Depth on Cutting Volume

Yasar and Yilmaz [20] reported that the cutting volume reaches a maximum at the optimal $s/p$ when using a pick cutter. Furthermore, the optimal $s/p$ did not exhibit a relationship with the strength of the rock. The same result was confirmed in cutting using the cutting tool of the precutting machine. The variation in the cutting volume with $s/p$ ratio is shown in Figure 11. The cutting volume reached a maximum at a certain spacing, and then decreased. It should be noted that the optimal $s/p$ at which the cutting volume reaches the maximum is the same, regardless of the penetration depth.

The cutting volume reached a maximum of approximately 5000 mm$^3$ when the $s/p$ ratio was 4 during cutting with a penetration depth of 3 mm (Figure 11a). Even when cutting with penetration depths of 6 mm and 9 mm, the cutting volume reached maximum values of approximately 13,000 mm$^3$ and 28,000 mm$^3$, respectively, with an $s/p$ ratio of 4 (Figure 11b,c). Furthermore, when $s/p$ exceeded 8 at a penetration depth of 3 mm, the cutting volume was constant at approximately 4000 mm$^3$ (Figure 11a). This result indicates that when $s/p$ becomes larger than 8, it is cut in the unrelieved mode, where the interaction between cutting does not occur. The details of the cutting volumes are listed in Tables 3–6.

### 3.2.3. Comparison of Measured Cutting Volume and Calculated Cutting Volume

The simple calculation method, one of the methods of measuring the cutting volume in the linear cutting test, is calculated from the cutting conditions and number of cuts, as shown in Equation (2).

$$V_{cal} = s \times p \times l \times (n - 1) \tag{2}$$

where $V_{cal}$ is the calculated cutting volume, $s$ is the spacing, $p$ is the penetration depth, $l$ is the cutting length, and $n$ is the number of cuts.

Because the simple calculation method is calculated under the assumption that all rocks between the cutters are removed, there is a problem that the cutting volume is overestimated if the cutter spacing increases more than the optimum spacing [10,32,33,38].

Figure 12 shows the variation in the ratio of the measured cutting volume to the calculated cutting volume according to $s/p$. This ratio was between 1.2 and 0.2, regardless of rock strength, and had a significant correlation with $s/p$ (R$^2$ = 0.641). When cutting with a disc cutter, this ratio reaches its maximum value, where the $s/p$ ratio is greater than 5 [10,32,33,38]. However, according to the regression curve in this study, this ratio reached its maximum value at an $s/p$ of approximately 1.5. This shows that when the spacing between the cuts is close to 1.5 times the penetration depth, the rock between the cutting lines is removed at or deeper than the penetration depth. However, when the spacing between cuts is greater than 1.5 times the penetration depth, ridges are expected

to be generated between the cutting lines. The details of the measured cutting volume, calculated cutting volume, and ratio of the measured to calculated cutting volumes are listed in Table 7.

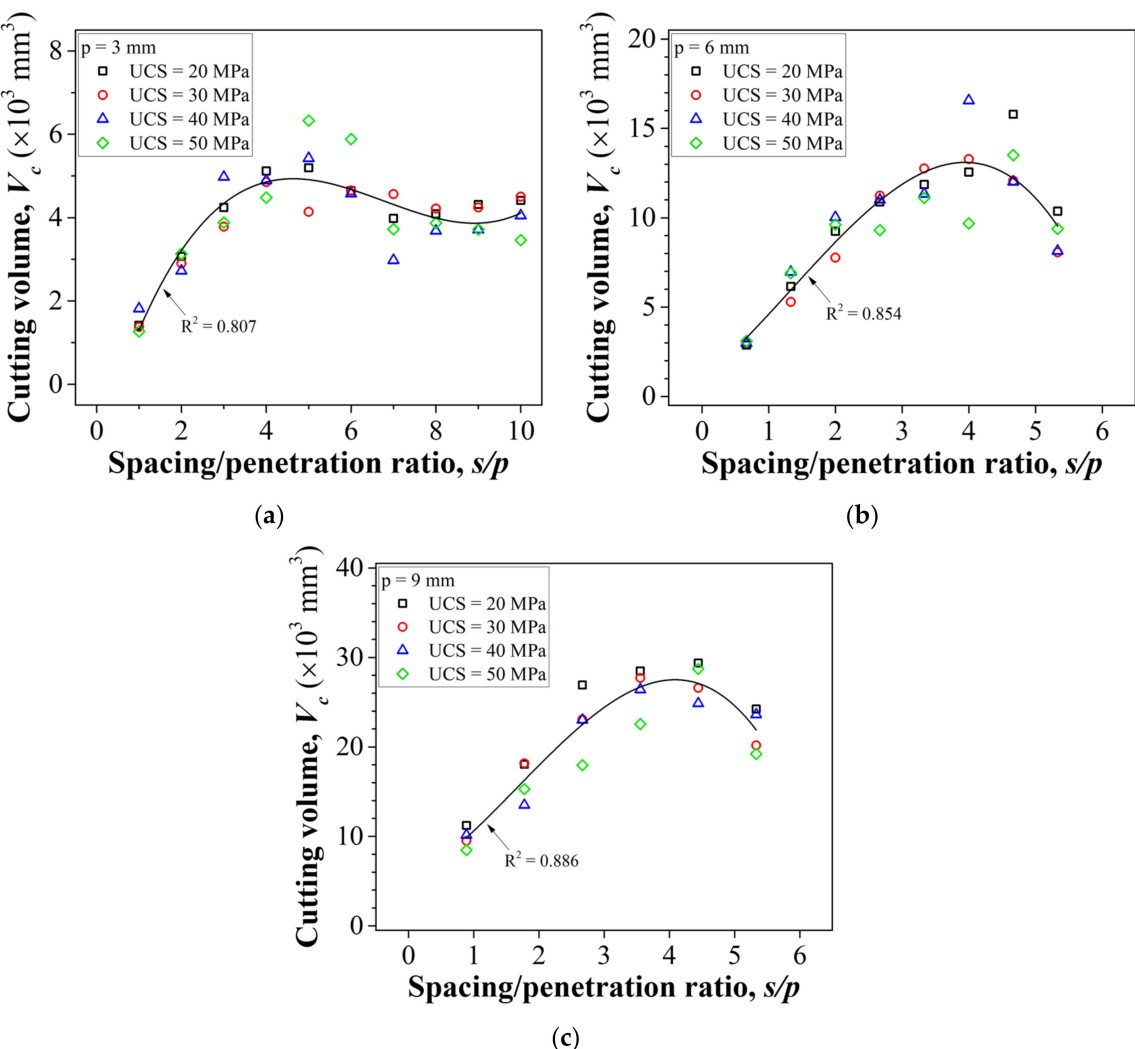

**Figure 11.** Relationships between cutting volume and $s/p$ ratio; (**a**) $p$ = 3 mm, (**b**) $p$ = 6 mm, and (**c**) $p$ = 9 mm.

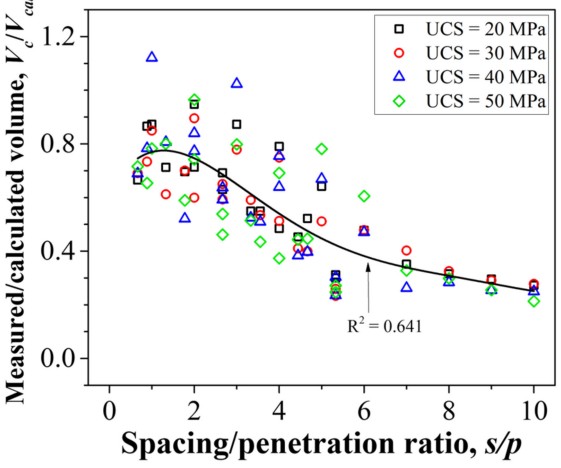

**Figure 12.** Relationship between $s/p$ ratio and measured to calculated cutting volume ratio.

**Table 7.** Measured and calculated cutting volume.

| $p$ (mm) | $s$ (mm) | $s/p$ | $l$ (mm) | $V_{cal}$ (mm³) | UCS (MPa) | | | | | | | |
|---|---|---|---|---|---|---|---|---|---|---|---|---|
| | | | | | 20 | | 30 | | 40 | | 50 | |
| | | | | | $V_c$ (mm³) | $V_c/V_{cal}$ | $V_c$ (mm³) | $V_c/V_{cal}$ | $V_c$ (mm³) | $V_c/V_{cal}$ | $V_c$ (mm³) | $V_c/V_{cal}$ |
| 3 | 3 | 1 | 180 | 1620 | 1414 | 0.87 | 1376 | 0.85 | 1376 | 0.85 | 1268 | 0.78 |
| | 6 | 2 | 180 | 3240 | 3070 | 0.95 | 2901 | 0.90 | 2901 | 0.90 | 3126 | 0.96 |
| | 9 | 3 | 180 | 4860 | 4242 | 0.87 | 3783 | 0.78 | 3783 | 0.78 | 3880 | 0.80 |
| | 12 | 4 | 180 | 6480 | 5121 | 0.79 | 4851 | 0.75 | 4851 | 0.75 | 4480 | 0.69 |
| | 15 | 5 | 180 | 8100 | 5195 | 0.64 | 4139 | 0.51 | 4139 | 0.51 | 6328 | 0.78 |
| | 18 | 6 | 180 | 9720 | 4646 | 0.48 | 4651 | 0.48 | 4651 | 0.48 | 5883 | 0.61 |
| | 21 | 7 | 180 | 11,340 | 3982 | 0.35 | 4563 | 0.40 | 4563 | 0.40 | 3720 | 0.33 |
| | 24 | 8 | 180 | 12,960 | 4091 | 0.32 | 4218 | 0.33 | 4218 | 0.33 | 3868 | 0.30 |
| | 27 | 9 | 180 | 14,580 | 4314 | 0.30 | 4245 | 0.29 | 4245 | 0.29 | 3716 | 0.25 |
| | 30 | 10 | 180 | 16,200 | 4411 | 0.27 | 4504 | 0.28 | 4504 | 0.28 | 3459 | 0.21 |
| 6 | 4 | 0.67 | 180 | 4320 | 2873 | 0.67 | 2985 | 0.69 | 2985 | 0.69 | 3093 | 0.72 |
| | 8 | 1.33 | 180 | 8640 | 6159 | 0.71 | 5288 | 0.61 | 5288 | 0.61 | 6915 | 0.80 |
| | 12 | 2 | 180 | 12,960 | 9241 | 0.71 | 7766 | 0.60 | 7766 | 0.60 | 9616 | 0.74 |
| | 16 | 2.67 | 180 | 17,280 | 10,885 | 0.63 | 11,237 | 0.65 | 11,237 | 0.65 | 9306 | 0.54 |
| | 20 | 3.33 | 180 | 21,600 | 11,860 | 0.55 | 12,761 | 0.59 | 12,761 | 0.59 | 11,111 | 0.51 |
| | 24 | 4 | 180 | 25,920 | 12,560 | 0.48 | 13,284 | 0.51 | 13,284 | 0.51 | 9680 | 0.37 |
| | 28 | 4.67 | 180 | 30,240 | 15,791 | 0.52 | 12,077 | 0.40 | 12,077 | 0.40 | 13,506 | 0.45 |
| | 32 | 5.33 | 180 | 34,560 | 10,366 | 0.30 | 8068 | 0.23 | 8068 | 0.23 | 9386 | 0.27 |
| 9 | 8 | 0.89 | 180 | 12,960 | 11,218 | 0.87 | 9507 | 0.73 | 9507 | 0.73 | 8471 | 0.65 |
| | 16 | 1.78 | 180 | 25,920 | 18,043 | 0.70 | 18,158 | 0.70 | 18,158 | 0.70 | 15,280 | 0.59 |
| | 24 | 2.67 | 180 | 38,880 | 26,906 | 0.69 | 23,066 | 0.59 | 23,066 | 0.59 | 17,956 | 0.46 |
| | 32 | 3.56 | 180 | 51,840 | 28,484 | 0.55 | 27,705 | 0.53 | 27,705 | 0.53 | 22,543 | 0.43 |
| | 40 | 4.44 | 180 | 64,800 | 29,367 | 0.45 | 26,586 | 0.41 | 26,586 | 0.41 | 28,715 | 0.44 |
| | 48 | 5.33 | 180 | 77,760 | 24,234 | 0.31 | 20,182 | 0.26 | 20,182 | 0.26 | 19,219 | 0.25 |

$p$: penetration depth, $s$: spacing, $l$: cutting length, $V_c$: measured cutting volume, $V_{cal}$: calculated cutting volume.

### 3.3. Analysis of the Characteristics of Ridge Formation

3.3.1. Effect of Penetration Depth on Ridge Formation

When cutting with a constant penetration depth, the rock between the cutting lines is completely removed up to the optimal spacing. However, if the spacing becomes larger than the optimal value, the rock between the cutting lines is not completely removed. If a ridge is formed between the cutting grooves, the cutting tool and cutter head of the tunnel excavation machine can be damaged [32]. Therefore, the characteristics of ridge formation during cutting using a pre-cutting machine were analyzed. The analysis was performed only on the rock specimen with a UCS of 20 MPa, and the cross-sectional profile was obtained by projecting a straight line onto the center of the damaged rock surface, as shown in Figure 13. The red part indicates the undamaged rock surface, and the gray part indicates the grooves generated from cutting.

Figures 14–16 show the cross-sectional profiles obtained from the cutting results of the rock specimens with a UCS of 20 MPa. The effect of penetration depth on ridge formation can be illustrated by comparing Figures 14h, 15f and 16c, which show cross-sectional profiles for a spacing of 24 mm. When cutting with a penetration depth of 3 mm, the ridge between the cutting grooves reached the height of the rock surface before cutting (Figure 14h). When cutting with a penetration depth of 6 mm, the ridge was formed to a height greater than half of the penetration depth (Figure 15f). However, the height of the ridge at a penetration depth of 9 mm is significantly smaller than the penetration depth (Figure 16c). In conclusion, the ridges between the cutting grooves decrease as the penetration depth increases.

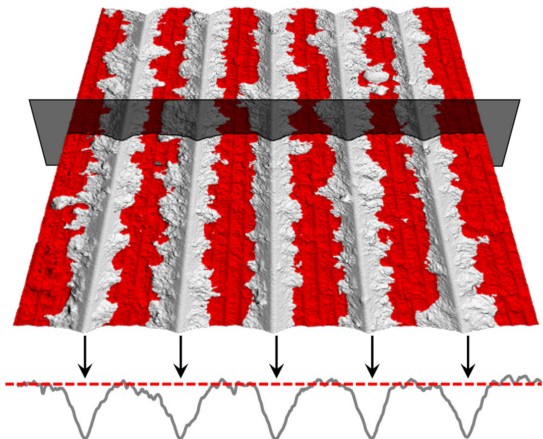

**Figure 13.** Example of obtaining a cross-sectional profile from a damaged rock surface.

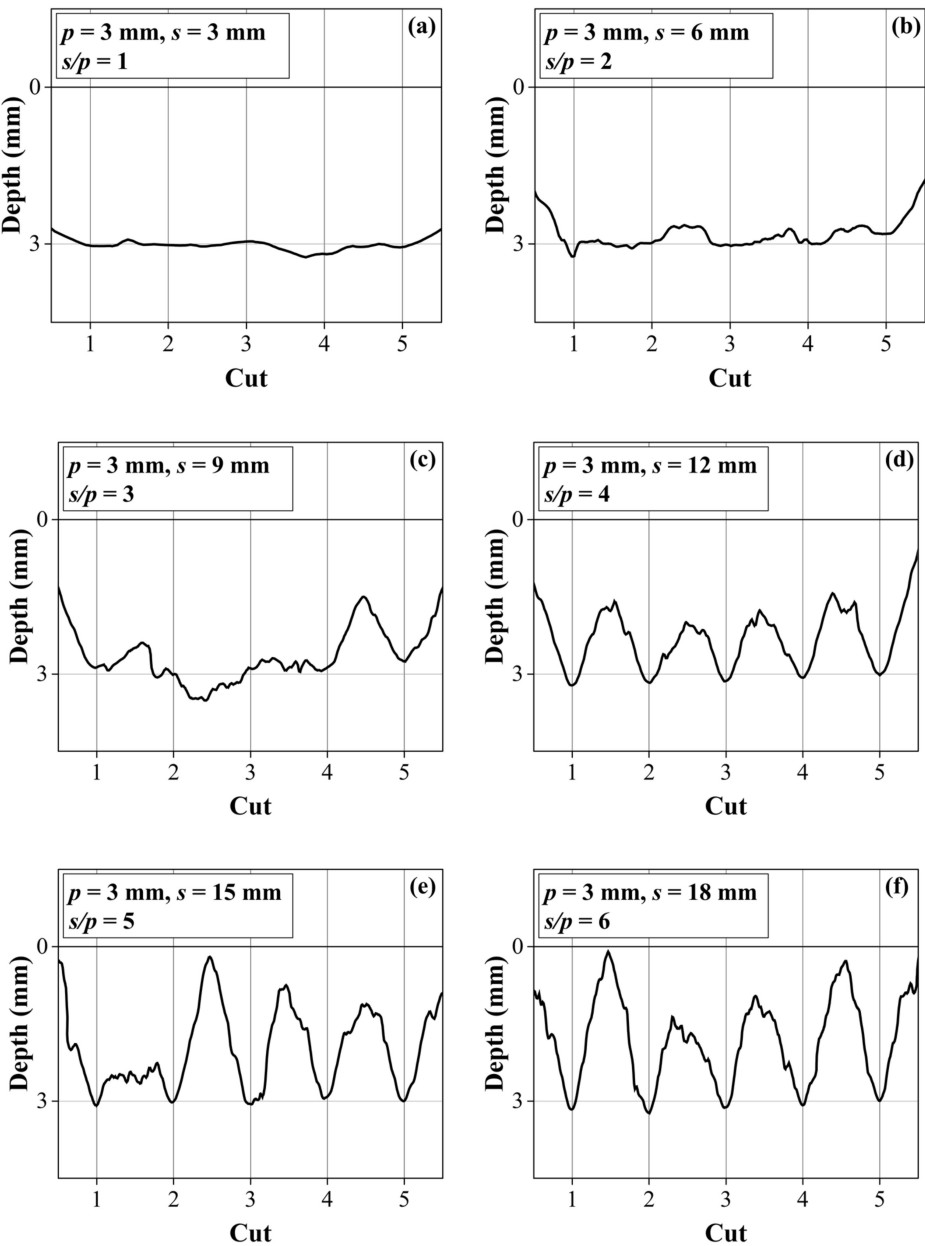

**Figure 14.** *Cont.*

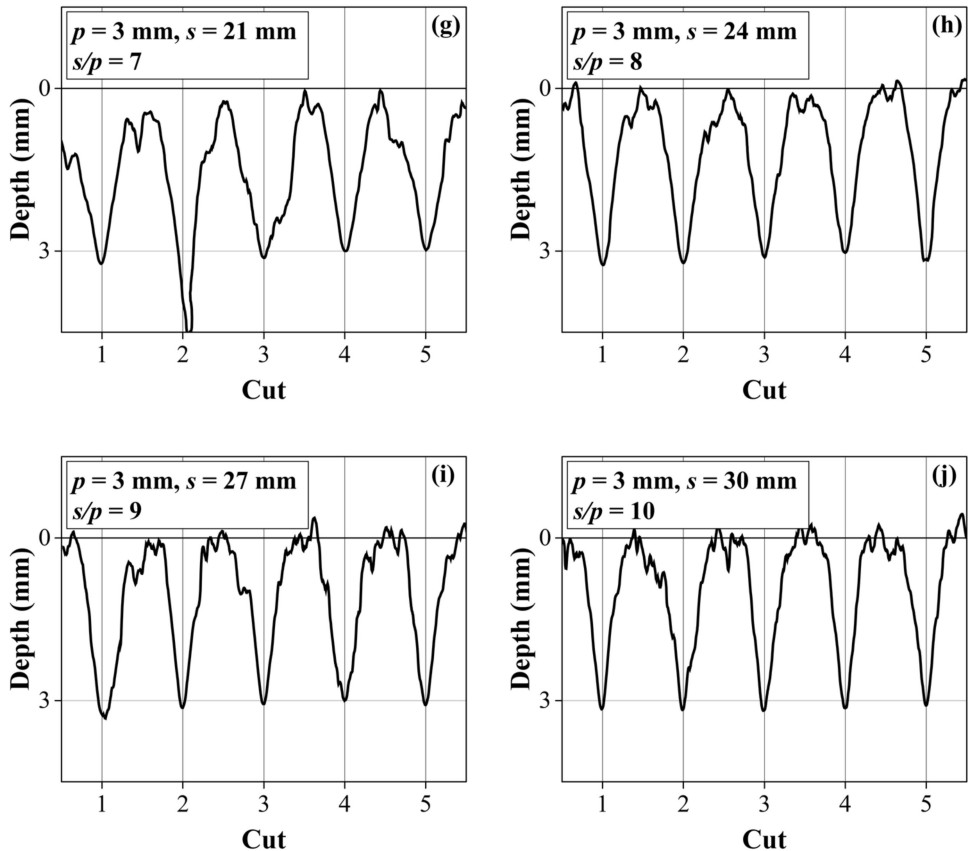

**Figure 14.** Cross-sectional profiles according to spacing when cutting a rock specimen of UCS 20 MPa with a penetration depth of 3 mm; (**a**) *s* = 3 mm, (**b**) *s* = 6 mm, (**c**) *s* = 9 mm, (**d**) *s* = 12 mm, (**e**) *s* = 15 mm, (**f**) *s* = 18 mm, (**g**) *s* = 21 mm, (**h**) *s* = 24 mm, (**i**) *s* = 27 mm, and (**j**) *s* = 30 mm.

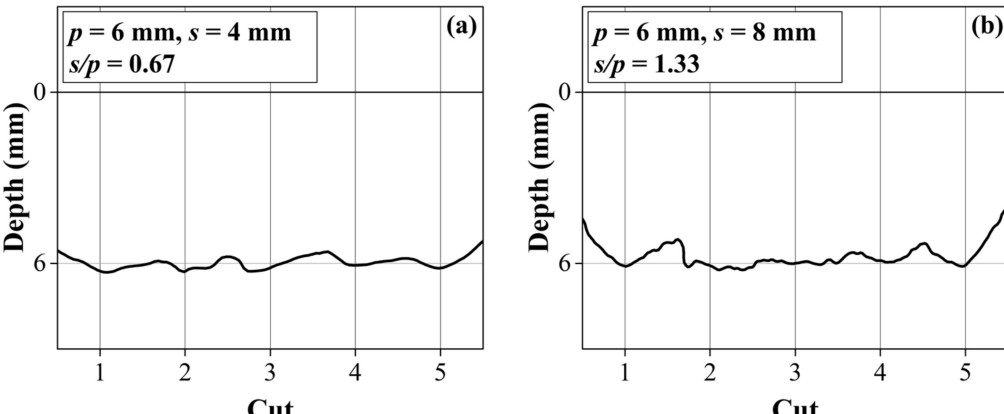

**Figure 15.** *Cont.*

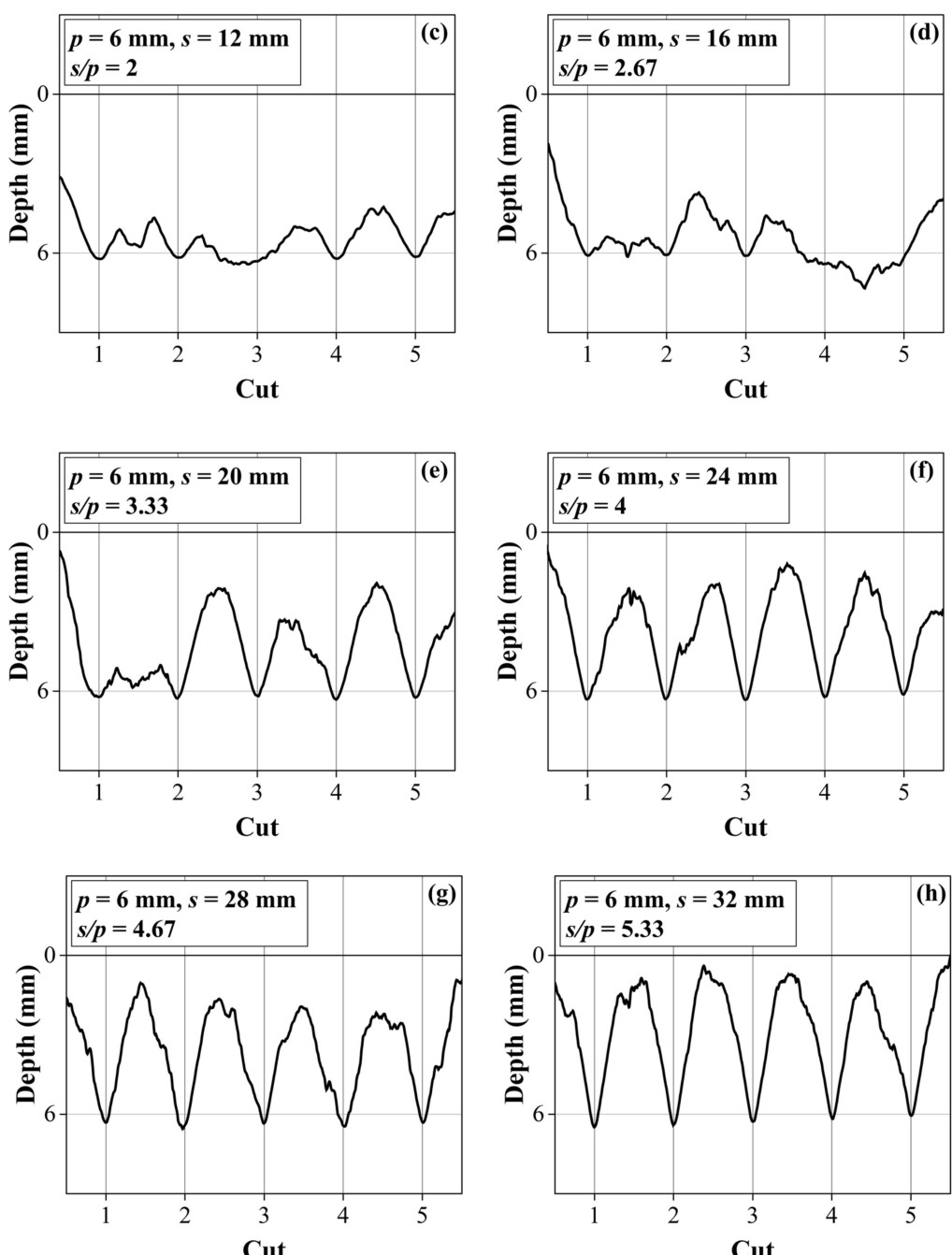

**Figure 15.** Cross-sectional profiles according to spacing when cutting a rock specimen of UCS 20 MPa with a penetration depth of 6 mm; (**a**) $s$ = 4 mm, (**b**) $s$ = 8 mm, (**c**) $s$ = 12 mm, (**d**) $s$ = 16 mm, (**e**) $s$ = 20 mm, (**f**) $s$ = 24 mm, (**g**) $s$ = 28 mm, and (**h**) $s$ = 32 mm.

3.3.2. Effect of Spacing on Ridge Formation

Figures 14–16 show that the height of the ridges rise with increasing spacing. In particular, ridge formation began when the cutting spacing was 6 mm with a penetration depth of 3 mm (Figure 14b). Moreover, in cutting with penetration depths of 6 mm and 9 mm, ridge formation began when the spacing was 8 mm and 16 mm, respectively (Figures 15b and 16b). This result supports the fact that the ridges are generated between the cutting grooves when the spacing exceeds 1.5 times the penetration depth, as mentioned in Section 3.2.3. In addition, the ridge reached the height of the rock surface before cutting, when the penetration depth and spacing were cut to 3 mm and 24 mm, respectively

(Figure 14h). This result proves that from the case that $s/p$ is 8, cutting is performed in an unrelieved mode in which no interaction occurs between cuts, as mentioned in Section 3.2.2.

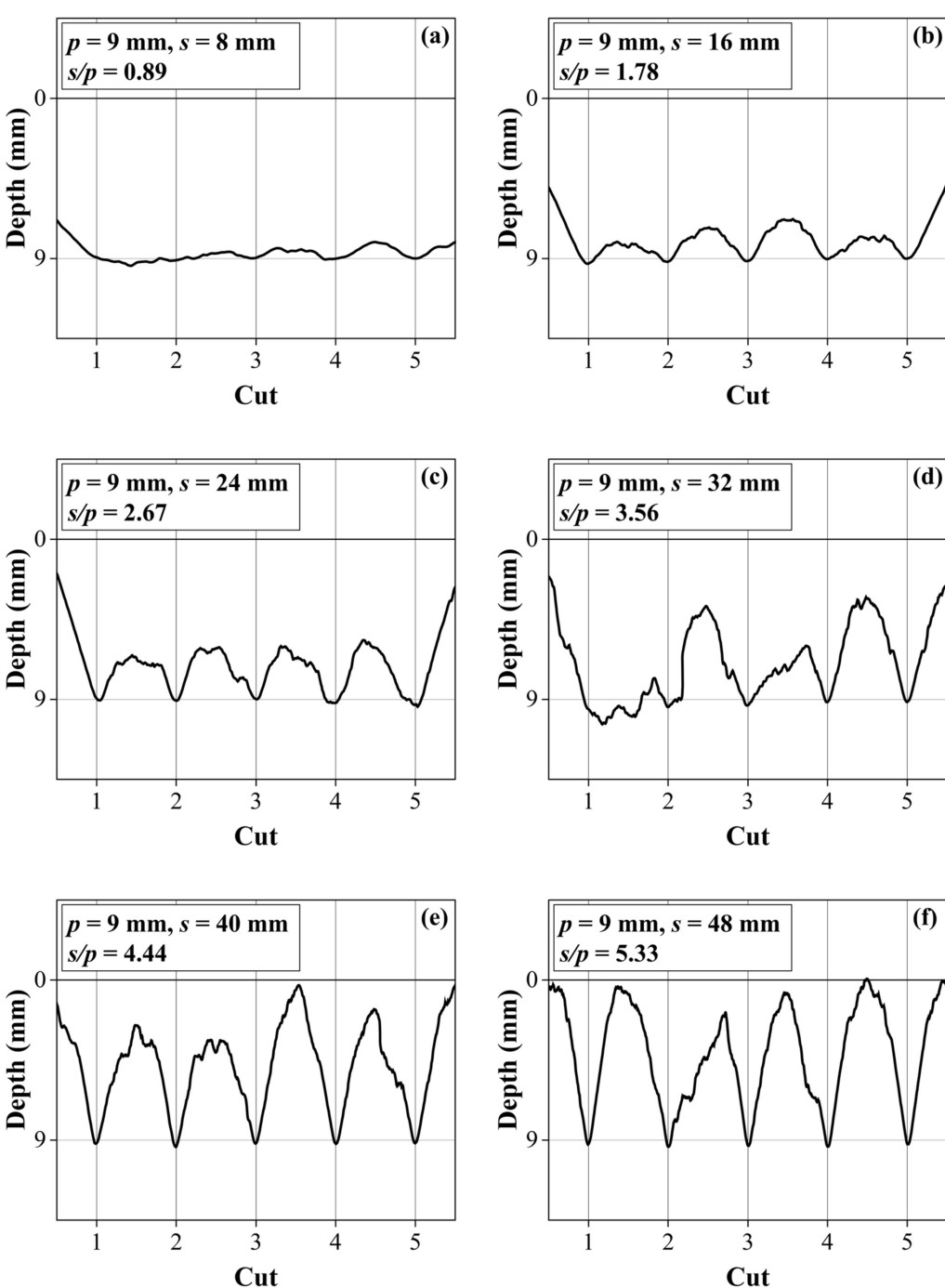

**Figure 16.** Cross-sectional profiles according to spacing when cutting a rock sample of UCS 20 MPa with a penetration depth of 9 mm; (**a**) $s$ = 8 mm, (**b**) $s$ = 16 mm, (**c**) $s$ = 24 mm, (**d**) $s$ = 32 mm, (**e**) $s$ = 40 mm, and (**f**) $s$ = 48 mm.

Figure 11 shows that the cutting volume reaches the maximum when $s/p$ is 4. However, Figures 14d and 15f show that the ridge formed was relatively high when $s/p$ was 4. The formation of the ridge is proof that excavation is inefficient. Furthermore, if the cutting volume reaches the maximum, the excavation efficiency obtained is different from the maximum. Therefore, it can be expected that the excavation efficiency depends on the cutting force instead of the cutting volume.

*3.4. Effect on Specific Energy*

3.4.1. Effect of Penetration Depth on Specific Energy

The cutting depth directly affects the cutting performance and efficiency of excavation machines. Regardless of the type of cutting tool, the specific energy decreases exponentially or as a power function as the penetration depth increases [10,19,30,32,39,40]. Bilgin et al. [14] stated that the specific energy in unrelieved mode cutting does not change effectively when the penetration depth is greater than 9–10 mm. Huang et al. [19] reported that the power value is not related to the type of rock in the power function relationship between the specific energy and penetration depth.

Figure 17 shows the relationships between the specific energy and penetration depth obtained from cutting using the pre-cutting machine cutting tool. At the same penetration depth, the greater the rock strength, the greater the specific energy. In addition, in all the rock specimens, the specific energy and penetration depth formed a power-function relationship. These results show that the effect of penetration depth on the specific energy of the pre-cutting machine is the same for other excavation machines. The details of the specific energy according to the penetration depth are listed in Table 8.

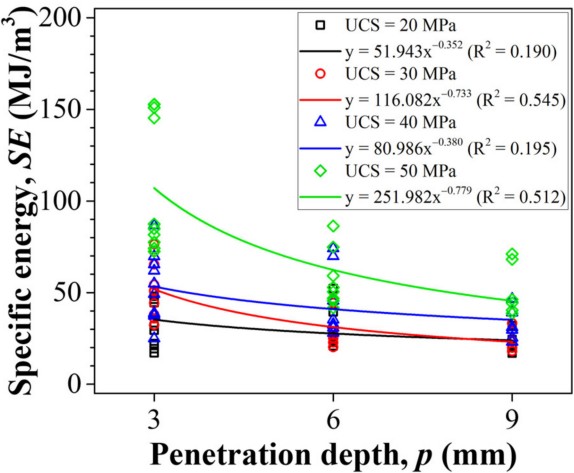

**Figure 17.** Relationships between penetration depth and specific energy.

**Table 8.** Specific energy according to cutting conditions.

| $p$ (mm) | $s$ (mm) | $s/p$ | UCS (MPa) | | | |
|---|---|---|---|---|---|---|
| | | | 20 | 30 | 40 | 50 |
| | | | $SE$ (MJ/m$^3$) | | | |
| 3 | 3 | 1 | 46.4 | 51.2 | 61.9 | 81.6 |
| | 6 | 2 | 22.2 | 36.0 | 38.8 | 72.5 |
| | 9 | 3 | 17.0 | 38.1 | 25.0 | 77.5 |
| | 12 | 4 | 19.3 | 32.3 | 38.2 | 84.8 |
| | 15 | 5 | 29.4 | 44.2 | 37.4 | 74.2 |
| | 18 | 6 | 31.8 | 50.7 | 49.2 | 87.4 |
| | 21 | 7 | 44.5 | 54.4 | 65.4 | 145.3 |
| | 24 | 8 | 44.6 | 65.4 | 54.9 | 152.7 |
| | 27 | 9 | 48.0 | 72.3 | 69.8 | 151.0 |
| | 30 | 10 | 44.6 | 76.4 | 86.7 | 152.8 |
| 6 | 4 | 0.67 | 52.2 | 44.4 | 74.2 | 86.3 |
| | 8 | 1.33 | 28.9 | 26.8 | 35.3 | 52.9 |
| | 12 | 2 | 23.8 | 20.2 | 32.4 | 45.5 |
| | 16 | 2.67 | 22.6 | 25.3 | 30.9 | 50.4 |
| | 20 | 3.33 | 25.1 | 22.8 | 32.8 | 41.7 |
| | 24 | 4 | 26.0 | 25.0 | 28.0 | 59.2 |
| | 28 | 4.67 | 20.9 | 28.4 | 45.6 | 46.9 |

**Table 8.** *Cont.*

| $p$ (mm) | $s$ (mm) | $s/p$ | UCS (MPa) | | | |
|---|---|---|---|---|---|---|
| | | | 20 | 30 | 40 | 50 |
| | | | $SE$ (MJ/m$^3$) | | | |
| | 32 | 5.33 | 39.4 | 46.3 | 69.9 | 74.8 |
| 9 | 8 | 0.89 | 31.9 | 33.4 | 46.5 | 71.1 |
| | 16 | 1.78 | 20.6 | 23.5 | 29.7 | 40.4 |
| | 24 | 2.67 | 16.9 | 18.0 | 23.2 | 46.3 |
| | 32 | 3.56 | 17.2 | 19.9 | 25.5 | 38.6 |
| | 40 | 4.44 | 19.9 | 23.3 | 31.1 | 44.4 |
| | 48 | 5.33 | 24.4 | 29.8 | 39.2 | 68.2 |

*p*: penetration depth, *s*: spacing, UCS: uniaxial compressive strength, *SE*: specific energy.

### 3.4.2. Effect of Spacing on Specific Energy

Spacing generally has a more dominant effect on specific energy compared to penetration depth [12]. If the spacing is too close, the specific energy is significantly high, and the cutting is inefficient because the rock is overcrushed. In addition, tool wear is high owing to the high friction between the rock and the tool. If the spacing is too wide, the specific energy is significantly high again, and the cutting is inefficient because cuts do not occur in the relieved mode where tensile fractures from adjacent cuts reach to form a rock chip. Furthermore, a ridge that might result in shock loads causing serious failure of the cutting tool or a stop of the machine is generated. The minimum specific energy is obtained using the optimal $s/p$ ratio. This represents the most efficient cutting conditions, largest rock chip, and minimum tool wear [14,41].

The relationships between the $s/p$ ratio and specific energy obtained in this study are shown in Figure 18. In the rock specimen of USC 50 Mpa, the optimal $s/p$ was 2.94 for cutting with a penetration depth of 3 mm, and 3.11 and 3.14 for penetration depths of 6 mm and 9 mm, respectively (Table 9). The optimal $s/p$ ratio at which the specific energy was minimized was almost the same, regardless of the penetration depth. For the other rock specimens, the optimal $s/p$ value was between 2 and 4, regardless of the penetration depth. This was consistent with several studies in which the optimal $s/p$ ratio of the pick cutter of the roadheader ranged from 2 to 5 [13,14,19,27]. Therefore, it can be expected that the cutting tool of the pre-cutting is similar to that of the pick cutter. The details of the specific energy according to spacing are listed in Table 8. The optimal $s/p$ ratio and the minimum specific energy are presented in Table 9.

**Table 9.** Optimal $s/p$ ratio and the resulting minimum specific energy.

| $p$ (mm) | UCS (MPa) | Opt. $s/p$ | Min. $SE$ (MJ/m$^3$) |
|---|---|---|---|
| 3 | 20 | 3.55 | 16.1 |
| | 30 | 3.48 | 35.2 |
| | 40 | 3.67 | 30.7 |
| | 50 | 2.94 | 68.7 |
| 6 | 20 | 3.26 | 20.1 |
| | 30 | 2.85 | 19.8 |
| | 40 | 2.99 | 25.2 |
| | 50 | 3.11 | 43.0 |
| 9 | 20 | 3.38 | 16.2 |
| | 30 | 3.21 | 18.5 |
| | 40 | 3.23 | 23.4 |
| | 50 | 3.14 | 37.3 |

*p*: penetration depth, UCS: uniaxial compressive strength, Opt. *s/p*: optimum spacing to penetration depth ratio, Min. *SE*: minimum specific energy.

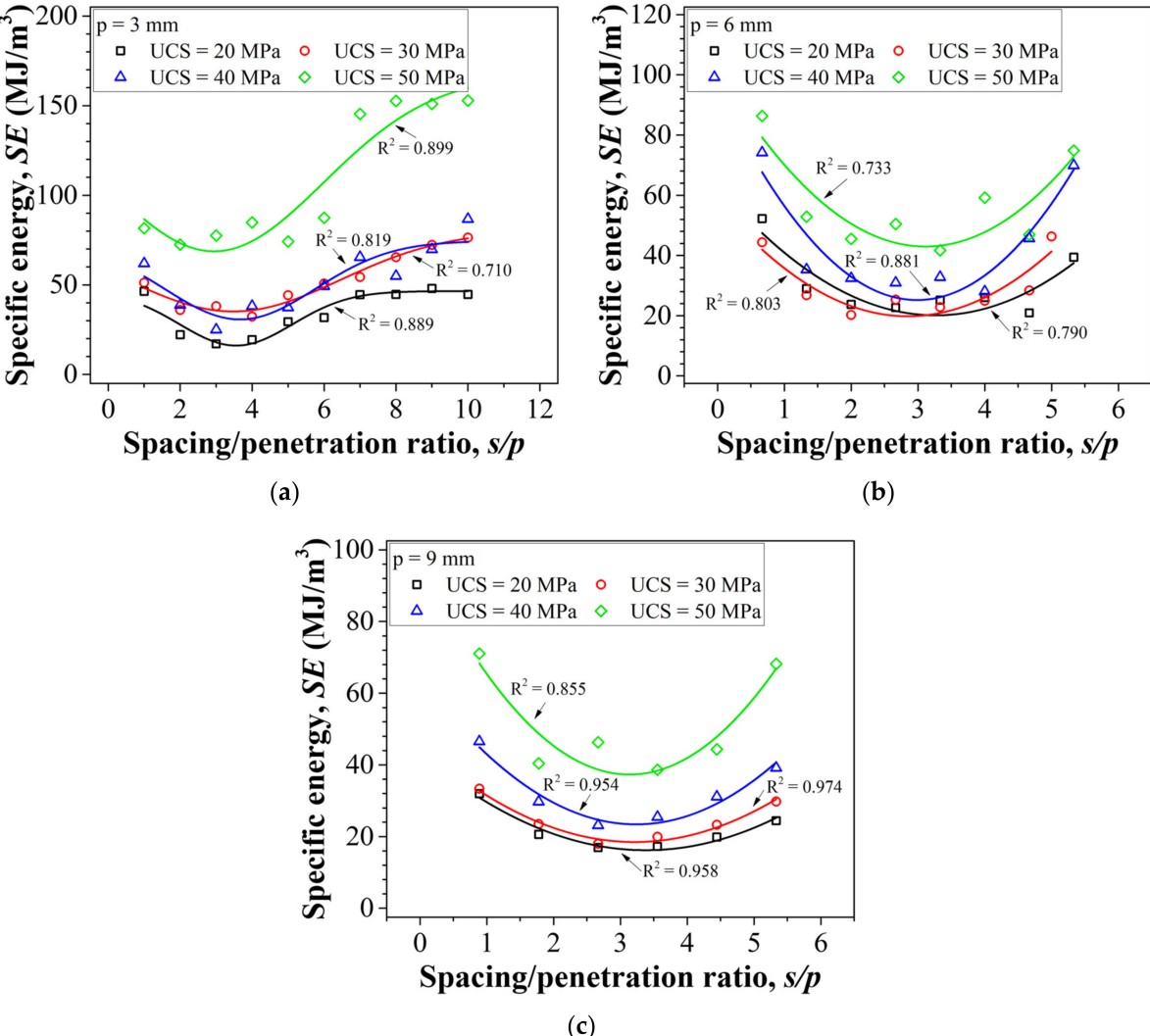

**Figure 18.** Relationships between *s/p* ratio and specific energy: (**a**) *p* = 3 mm, (**b**) *p* = 6 mm, and (**c**) *p* = 9 mm.

## 4. Conclusions

In this study, a series of linear cutting tests was performed using the cutting tool of a pre-cutting machine. Regression analyses were performed according to the tool forces, cutting volumes, and specific energies recorded from rock cutting. Furthermore, the ridge formation characteristics between the cutting grooves were analyzed from the cross-sectional profile of the rock after cutting obtained from 3D scanning. The conclusions of this study based on the test results are as follows.

1.  The tool forces increased as the penetration depth and spacing increased, similar to TBMs and roadheaders. However, information for predicting optimal cutting conditions can be obtained from the ratio of tool forces. The ratio of the peak cutting force to the mean cutting force was the maximum near 2 for *s/p*, and the ratio of the cutting force to the normal force was also the maximum near 2 for *s/p*. In conclusion, when *s/p* is near 2, larger rock chips are obtained, and the wear of the tool is minimized.

2.  The cutting volume increased in the power relationship as the penetration depth increased, and was not affected by the rock strength. This volume reached a maximum at a specific *s/p*, and the specific *s/p* was the same, regardless of the penetration depth. This result is similar to the case of the TBMs and roadheaders. Additionally, the cutting

volume was constant as $s/p$ became greater than 8, and the ratio of the measured cutting volume to the calculated cutting volume reached a maximum when $s/p$ was close to 1.5. This result shows that ridges start to form between the cutting grooves as $s/p$ becomes larger than 1.5, and the cutting is performed in the unrelieved mode if it becomes larger than 8.

3.  The ridge between the cutting grooves decreased as the penetration depth increased and increased with spacing. In particular, when $s/p$ was 1.5, ridges started to form, and as $s/p$ became larger than 8, the ridges reached the height of the rock surface before cutting. This supports the results obtained from the analysis of cutting volume.

4.  The specific energy decreased in a power relationship as the penetration depth increased, similar to the case of the TBMs and roadheaders. In addition, the specific energy reached a minimum when $s/p$ was between 2 and 4, regardless of the penetration depth and rock strength, similar to that of a roadheader using a drag tool.

**Author Contributions:** H.-e.K. proposed the concept of research, developed the study, and conducted the experiment. S.-g.H. and H.R. contributed to reviewing the final paper and made recommendations for revision. H.-k.Y. supervised the study and provided important suggestions. All authors have read and agreed to the published version of the manuscript.

**Funding:** This work was supported by the National Research Foundation of Korea Grant funded by the Korean Government (NRF-2019R1A2C2003636).

**Institutional Review Board Statement:** Not applicable.

**Informed Consent Statement:** Not applicable.

**Data Availability Statement:** Not applicable.

**Conflicts of Interest:** The authors declare no conflict of interest.

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
