# Peer review of "Analysis of Mechanical Excavation Characteristics by Pre-Cutting Machine Based on Linear Cutting Tests"

_applsci, doi:10.3390/app13021205_

Round 1

Reviewer 1 Report

The study is very thorough, however please address the below points prior to acceptance.

- Add more references relevant to the linear cutting. Instead of equipment focus more the literature of linear cutting and related problems.

- The literature review should point out at the research gaps, then your study should justify that it is addressing the research gap. Please adjust introduction to convey the same idea. 

- Originality of the work needs to be highlighted in manuscript and abstract as well.

- Abstract should have some qualitative data as well from this work.

- Why the cutting tool with specific geometry (rake angle etc.) was selected? 

- Is there any ISO specification for the tool?

- What standards are used in the material testing? Generally, ASTM standards are followed. 

- There is a need to connect the discussion of forces data with the rock fracture mechanics. This is required to capture the physics involved. Please provide the discussion and adjust conclusions accordingly.

Reviewer 2 Report

1. The first paragraph from Section 2.1:

It’s not a strong rule about the use of a certain type of rock cutting tools for partial-face or full-face excavation. Thus, the paragraph should be re-written or excluded.

2. String 99:

What are the reasons to include the tool’s shape from a previous work?

3. Authors choose TBMs and roadheaders as a reference point for their study. Although the pre-cutting method is in the use of chain-saw-shaped cutter head. So, the authors should explain why they do not prefer circular saw blades or similar machines and related technologies.

4. Strings 192-198:

What is the purpose of this analysis? The whole paragraph seems like an attempt to increase manuscript’s volume.

5. String 192-193:

Do authors think that difference for the values of R2, which is about 2,5%, is enough for making some conclusions? What conclusions?

6. Strings 234-235:

The value of R2 = 0,495 does not mean “significant correlation”. Rather, this could be interpreted as unsatisfactory correlation.

7. Strings 399-402:

Why patterns from rock cutting by conical picks are used for description and explanation of a process with the use of another tool?

8. String 458:

Typing mistake – should be “roadheaders” instead of “load headers”.

Reviewer 3 Report

This research paper focused on analysis of mechanical excavation characteristics by pre-cut ting machine based on linear cutting tests. The research methodology, experiments design and results analysis are reasonable. This manuscript was written well. I suggested to publish this work after minor revision. The detailed comments were shown as follows.

1. In the abstract, the result tense should be past tense. The results showed……

2. This study can provide useful information on, the “on” is suggested to revised to “for”.

3. Some texts are marked as yellow color, which are need to be concerned.

4. In table 1, the significant digits of a number should be consistent.

5. Line 142, cutting force (Fc) is not correct. Cutting force includes all the force in different direction.

6. Line 159, This series should be revised to “These series”.

7. Line 164, Because the cutting speed does not affect the cutting performance. Why?

8. For the cutting force and specific energy, the following works are suggested to cite. https://doi.org/10.3390/machines10050331

https://doi.org/10.1080/17480272.2022.2049867

9. “Tool Forces” is suggested to revised to “Cutting forces”.

Round 2

Reviewer 1 Report

It can be accepted.